# Prevalence of the depression among heart failure patients in Ethiopia, 2024: A systematic review and meta-analysis

**Birhaneslasie Gebeyehu Yazew**[1]*, **Workineh Tamir Shitu**[2‡], **Mohammed Hassen Salih**[3‡], **Zewdu Bishaw Aynalem**[1‡], **Haile Workye Agazhu**[1‡], **Daniel Adane Endalew**[4‡]

1 Department of Nursing, College of Medicine and Health Science, Injibara University, Injibara, Ethiopia, 2 Department of Medical Laboratory Science, College of Medicine and Health Sciences, Injibara University, Injibara, Ethiopia, 3 Department of Medical Nursing, School of Nursing, College of Medicine and Health Sciences, University of Gondar, Gondar, Ethiopia, 4 Department of Midwifery, College of Medicine and Health Sciences, Injibara University, Injibara, Ethiopia

‡ These authors also contributed equally to this work.
* kassish6@gmail.com

## Abstract

### Background

Heart failure (HF) is a major clinical condition contributing to high morbidity, mortality, and healthcare burden. Depression is increasingly recognized as a nontraditional risk factor for HF. However, data on its prevalence among HF patients in Ethiopia remain limited. This study aimed to estimate the pooled prevalence of depression in Ethiopian HF patients.

### Methods

The study followed the Preferred Reporting Items for Systematic Reviews and Meta-Analyses guidelines, using data abstraction from various electronic sources (PubMed, Web of Science, Google Scholar, Scopus, Science direct, African journal, and online University repositories studies). Studies reporting the prevalence of depression among heart failure patients found until 28th November, 2024 were included. Analysis was conducted using STATA version 17 software, with assessment of heterogeneity and publication bias. A random-effects meta-analysis model was used to estimate the pooled prevalence of depression.

### Results

This study revealing a pooled prevalence of depression among Heart failure patients in Ethiopia is 43.93%. Subgroup analyses based on region, type of institution, and sampling method showed different prevalence rates: the Southern Nations, Nationalities, and Peoples' Region had the highest rate at 60.13%, while Addis Ababa had the lowest at 35.18%. In terms of study institution types, teaching hospitals reported

**Data availability statement:** All relevant data are within the paper and its Supporting Information files.

**Funding:** The author(s) received no specific funding for this work.

**Competing interests:** The authors have declared that no competing interests exist.

the highest prevalence at 46.81%, whereas referral hospitals showed the lowest rate at 31.05%. When considering sampling techniques, consecutive sampling yielded the highest prevalence at 55.09%, compared to just 15.40% for systematic random sampling. The analysis indicated a publication bias (p = 0.003), which warranted the use of trim and fill methods.

## Conclusion and recommendations

The rate of depression among heart failure patients in Ethiopia is notably high, highlighting the necessity for targeted interventions from the Ministry of Health to tackle this concern. It is essential to create multi-sectorial strategies that offer context-specific solutions, such as rehabilitation programs, to help reduce depression in heart failure patients. The review protocol was registered in PROSPERO ID: CRD42023405077.

## Introduction

Heart failure (HF) is a multifaceted clinical condition [1] that impacts over 64 million individuals worldwide, at various stages [2]. It is a leading cause of morbidity and mortality, resulting in high rates of hospital readmissions and significant healthcare expenses [3]. Many patients face psychological challenges stemming from the chronic nature of the illness, medication regimens, and bleak prognoses. Depression, in particular, is acknowledged recognized as a significant yet unconventional risk factor for cardiovascular diseases, including heart failure [4], and linked to high mortality rates [5]. Approximately 21.6% of those with heart failure suffer from depression [6], leading to a 30% rise in healthcare costs [7].

Recent literature reviews from around the globe have shown that the prevalence of depressive symptoms in individuals with heart failure (HF) varies significantly. Research indicates a strong link between depression and HF, with both conditions often occurring together [8]. The worldwide prevalence of depression among HF patients can reach up to 41.9% [9], while in China, this figure rises to 51% [10]. In developing nations such as Ethiopia, depression represents about 6.5% of the overall disease burden, making it the most significant contributor among mental health disorders [11].

Although the significance of addressing depression in heart failure patients is clear, there have been relatively few studies in Ethiopia that document the high prevalence of depression alongside heart failure. While depressive symptoms are acknowledged as a major complication of heart failure in the country, research findings have been inconsistent, with prevalence rates ranging from 11.1% to 57.5% [12–22]. Therefore, the primary objective of this review is to determine the overall pooled prevalence of depression among heart failure patients in Ethiopia by analyzing available studies. The results of this review will provide valuable insights for policymakers and program planners in developing effective interventions to reduce depression among heart failure patients in the country. Additionally, the findings will be beneficial for clinicians and future researchers working in related fields.

## Main text

### Materials and methods

**Study protocol registration.** This study had submitted and registered in the International Prospective Register of Systemic Review (PROSPERO) database with the protocol number CRD42023405077.

**Search strategy.** A systematic literature search had made to identify information through PubMed, Web of Science, Google Scholar, Scopus, Science direct, African journal, and online University repositories. Based on Preferred Reporting Items for Systematic Reviews and Meta-Analyses (PRISMA) statement [23]. The consequent keywords were had used on behalf of the PubMed database searching using balloon operators. Such as; "prevalence of depression among Heart failure in Ethiopia", "Depression" OR "depress" OR "Mental disorder" OR "Mood disorder" AND "Heart failure"OR "congestive heart failure" AND "Ethiopia". Moreover, we observed the reference lists of published studies to recognize more articles. Further, gray literature and reference lists of relevant articles were also retrieved to find extra studies. We had conducted a search from 1st January up to 24th March 2023, and then updated 1st October up to 28th November, 2024. The studies were retrieved through PubMed = 19, Web of Science = 124, Google Scholar = 178, Scopus = 15, Science direct = 275, African journal = 548, and online University repositories = 2. All published and unpublished articles found until 28th November 2024 were included in this review. The detailed search strategies are found in the supporting information (S1 Table).

**Eligibility criteria.** Articles meeting the specified criteria were reviewed for eligibility, including studies conducted in Ethiopia, published in peer-reviewed journals or gray literature, using observational cross-sectional study designs, focusing on the prevalence of depression among individuals with heart failure. Only articles in English were considered, and EndNote (version X7) software was used for reference management. Studies that provided clear descriptions of participants, including the number of participants tested for depression and the prevalence of depression among heart failure cases in specific regions of Ethiopia, were included. Review articles, conference abstracts, sentinel reports, and case reports were excluded from the review process.

**Outcome measures.** Ten studies employed the Patient Health Questionnaire (PHQ-9) to assess depression, defining an individual as depressed if their score was ≥ 10, with higher scores reflecting more severe depression [24]. In contrast, one study utilized the Beck Depression Inventory and classified patients with moderate to severe scores as having depression [25]. Additionally, heart failure was diagnosed based on the heart's inability to pump blood effectively, as indicated by symptoms and signs according to the Framingham criteria or a reduced ejection fraction of less than 40% [26,27]. The prevalence of depression among heart failure patients was determined by dividing the number of those diagnosed with both conditions by the total number of heart failure patients attending cardiac clinic follow-ups.

**Research questions.** 1.What is the estimated pooled prevalence of depression among HF in Ethiopia? The primary outcome of this review was measuring prevalence of depression among HF in Ethiopia.

Study selection, quality assessment, and data extraction Literature's using searching terms and titles had searched. Among them irrelevant and commentary abstracts had removed. That was beside to those removed due to duplication. Pertinent full texts of the articles had reviewed for eligibility by two reviewers (BGY and ZBA). Joanna Brigg's Institute (JBI) quality assessment checklist had used to check quality and this tool has nine indicators. The quality of each included article was classified as higher (>80%), moderate (65%−80%), or low (< 60%). For prevalence studies, which had scored greater than or equal to 60% in checklists were [28,29] included (S2 Table). Using Microsoft Excel 2016, data had extracted under the following information: authors name, year of publication, region, study design, quality, sample size, type of Hospital, sampling technique and prevalence rate of depression. Four authors (BGY, ZBA, HWA, and MHS) were conducted the data extraction. Any inconsistencies of the idea had resolved with the presence of two authors (DAE and WTS).

**Data analysis and synthesis.** The data was inputted into the computer via a STATA v.17 command window for analysis. A random-effects model was utilized to estimate the pooled prevalence of depression, with a focus on managing

heterogeneity among studies. The Dorsmanin and Laird method was employed to address heterogeneity [30,31], a common technique in random-effects meta-analysis [32]. Data manipulation and statistical analyses were conducted using STATA software, version 17.

The presence of heterogeneity among studies was assessed using $I^2$ test statistics, which quantifies the percentage of variation attributed to heterogeneity [33]. A high $I^2$ value (>75%) indicated significant heterogeneity, leading to the use of a random-effects model with 95% confidence intervals for analysis. This approach was chosen to account for observed variability and adjust for the substantial heterogeneity present in the data [34].

Moreover, the presence of heterogeneity had also evaluated by subgroup analysis. Such as by study area/region, types of institution which was conducted. Finally, meta-regression had done to overcome heterogeneity.

The inspection of publication bias was conducted using a funnel plot. Asymmetry of the funnel plot is an indicator of publication bias. Egger's tests was also conducted to check the potential publication bias. A p-value of less than 0.05 had used to declare the statistical significance of publication bias [35,36]. Additionally, a sensitivity analysis was also done. Because, to review whether the pooled prevalence estimates had influenced by individual studies or not.

## Results

The flow chart depicts the study selection process. Initially, 1161 studies were identified through a literature search. After eliminating 96 records prior to screening, 1044 irrelevant studies were excluded based on their titles and abstracts. Of the remaining 21 articles, 2 were found to be irrelevant and categorized as commentary literature. The full texts of the 19 remaining articles were evaluated, resulting in the exclusion of an additional 8 studies. The reasons for exclusion included lack of reporting on the outcome of interest and studies conducted outside of Ethiopia (S3 Table). Ultimately, 11 distinct studies met the eligibility criteria and were included in the final analysis (Fig 1).

**Characteristics of included studies.** A total of 11 studies with 3807 participants included in this systematic review had summarized in Table 1. The studies had conducted from 2019 to 2024 in different regions of the country. Among 11 studies, four of them [12,13,18,22] were in the Addis Ababa (AA), three of them [14,15,20] were in the Amhara region, three studies of them [16,17,21] were in Oromia region, and one [19] study of them was in Southern Nation Nationalities and Peoples' of Region (SNNPR). However, the area where the study was conducted (SNNPR) is now reclassified and incorporated into the newly established Central Ethiopia region. The least [13] and most [12] study sample size was from AA. Besides, all studies enrolled in this systematic review were cross-sectional studies (Table1).

**Prevalence of depression among adults with Heart failure (Systematic review).** The combined prevalence of depression among heart failure patients varied significantly between studies when analyzed using both fixed-effect and random-effects models. The reviewed prevalence of depression among HF was categorized by severity (mild, moderate, moderate-to-severe, and severe), which was 24.94%, 25.2%, 17.1%, and 26.81% respectively. Thus, the overall pooled prevalence of depression reported by the eleven studies was 43.93% (95% CI: 30.71%−57.15%) using the random-effects model, with a high level of heterogeneity ($I^2 = 98.9\%$, $p \leq 0.001$) (Fig 2).

Subgroup analyses were performed based on the study location, type of hospital, and sampling method used to identify potential sources of variability. Among the eleven studies, the highest estimated prevalence of depression in heart failure patients was recorded in the Southern Nations, Nationalities, and Peoples' Region (SNNPR) at 60.80% (95% CI: 54.62, 66.98), with no observed heterogeneity and this might be due to single study. Conversely, the lowest prevalence was noted in Addis Ababa (AA) at 35.18% (95% CI: 11.36, 59.00), exhibiting high heterogeneity ($I^2 = 99.1\%$, $p \leq 0.001$) (Fig 3).

Further subgroup analyses categorized by the type of healthcare institution indicated that the prevalence of depression was highest in teaching hospitals at 46.81% (95% CI 33.29, 60.32), with significant heterogeneity ($I^2 = 98.5\%$, $p \leq 0.01$), while referral hospitals reported the lowest rate at 31.05% (95% CI: 8.14, 70.25), also showing considerable heterogeneity ($I^2 = 99.5\%$, $p \leq 0.01$) (Fig 4).

**PRISMA 2020 flow diagram**

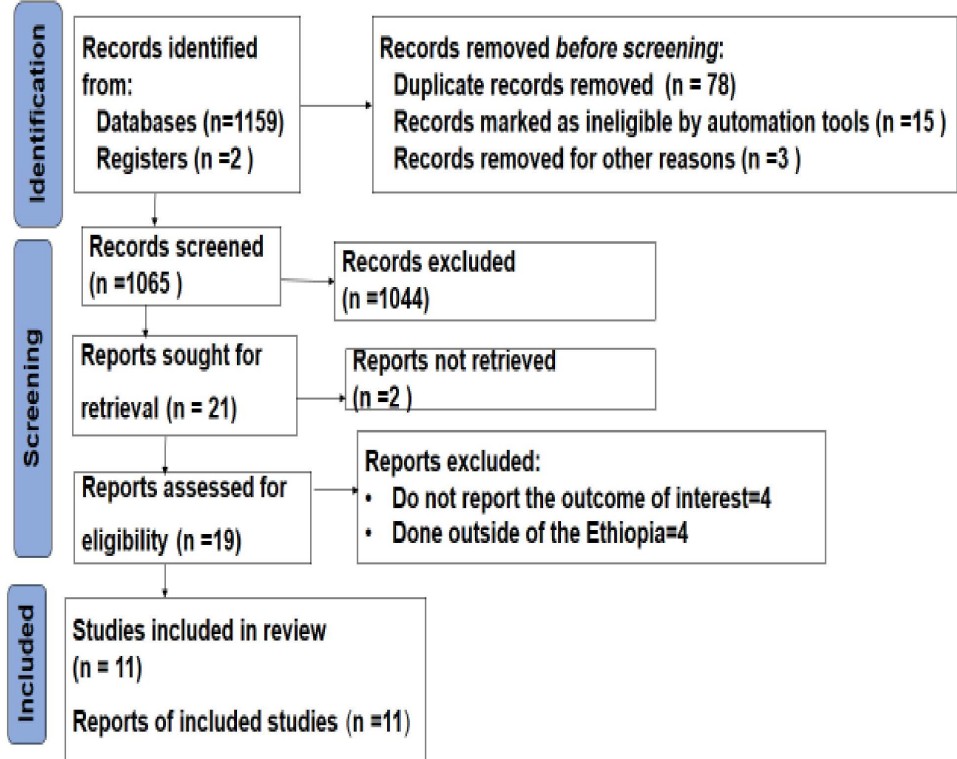

**Fig 1. Flow chart of study selection for systematic review of prevalence of Depression among Heart failure patients in Ethiopia, 2024.**

**Table 1. Characteristics of studies included in systematic review of the prevalence of Depression among Heart Failure patients in Ethiopia, 2024.**

| S.no | Author/ References | Publication Year | Typ. Hospital | Quality | Region | Study design | Sampling technique | Sample size | Prevalence (95% CI) |
|---|---|---|---|---|---|---|---|---|---|
| 1 | Alemayehu K et al [12] | 2022 | TH | 8 | AA | CS | SRS | 424 | 56.1(51.4,61) |
| 2 | Surafel W et al [13] | 2022 | TH | 7 | AA | CS | SRS | 164 | 12.7(7.6,17.8) |
| 3 | Afework E et al [14] | 2020 | RH | 8 | Amhara | CS | Stratified RS | 384 | 11.1(7.96, 14.3) |
| 4 | KG Yazew et al [15] | 2019 | RH | 7 | Amhara | CS | SRS | 403 | 51.1(46.2,56) |
| 5 | Halima et al [16] | 2019 | TH | 9 | Oromia | CS | SRS | 284 | 52.8 (47 59 ) |
| 6 | Belete et al [17] | 2019 | TH | 8 | Oromia | CS | SRS | 339 | 57.5(52.3, 63) |
| 7 | Henok et al [18] | 2024 | TH | 9 | AA | CS | Consecutive | 383 | 56.6(51.63, 61.56) |
| 8 | Ermias et al [19] | 2024 | TH | 8 | SNNPR | CS | SRS | 240 | 60.8(54.62, 66.97) |
| 9 | Tihitna et al [20] | 2024 | TH | 9 | Amhare | CS | Consecutive | 370 | 53.51(48.42,58.59) |
| 10 | Almaze et al [21] | 2022 | TH | 7 | Oromia | CS | SRS | 420 | 56.2(51.45, 60.94) |
| 11 | Tegegne et al [22] | 2021 | TH | 9 | AA | CS | Systematic RS | 396 | 15.4(11.84, 18.95) |

**Note: – TH-**Teaching Hospital, **RH-**Referral Hospital, **AA-**Addis Ababa, **CS-**Cross-sectional, **SNNPR-**Southern Nation Nationalities and Peoples' Region, **SRS-**Systematic Random Sampling

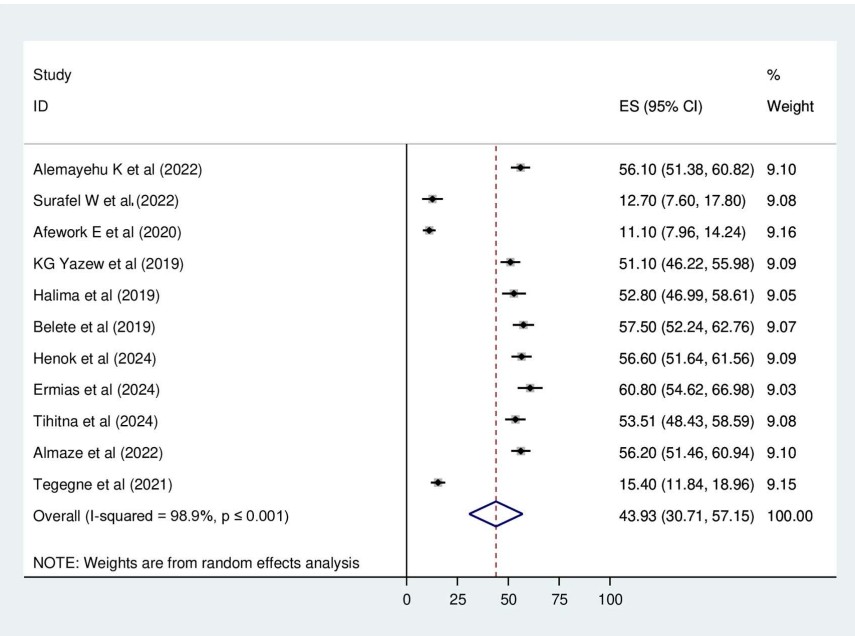

**Fig 2. Forest plot showing the pooled prevalence of Depression among Heart failure patients in Ethiopia, 2024.**

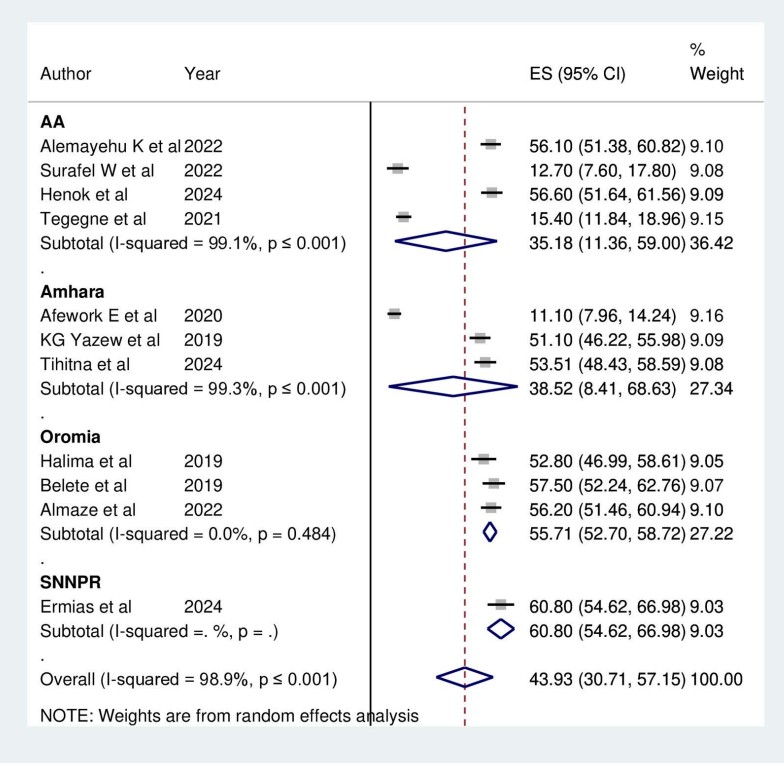

**Fig 3. Subgroup analysis by regions on the prevalence of Depression among Heart failure patients in Ethiopia, 2024.**

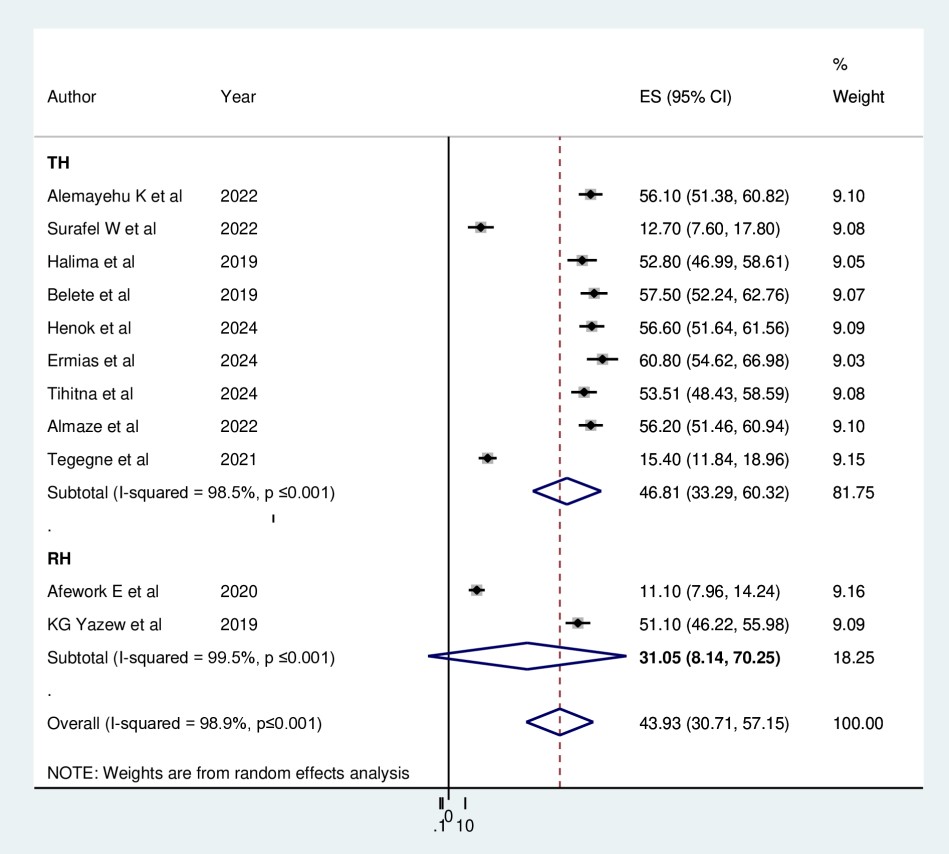

**Fig 4. Subgroup analysis by types of Hospital on the prevalence of Depression among Heart failure patients in Ethiopia, 2024.**

Lastly, subgroup analysis by sampling method revealed that the prevalence of depression was highest with consecutive sampling at 55.09% (95% CI: 51.54, 58.64), showing no heterogeneity ($I^2 = 0.0\%$, p = 0.394), while systematic random sampling yielded the lowest prevalence at 15.40% (95% CI: 11.84, 18.96). Although consecutive sampling and systematic random sampling indicated potential reasons for the observed disparities, the simple random sampling technique demonstrated significant heterogeneity ($I^2 = 97.5\%$, p ≤ 0.001) (Fig 5). Therefore, the results of the subgroup analysis suggest that the identified sources of heterogeneity were not attributed to the region, institution type, or sampling method.

**Funnel plot of depression heterogeneity investigation.** The systematic review revealed significant variability among the studies included, as indicated by the Cochrane Q-test (p < 0.001) and the $I^2$ test ($I^2 = 98.9\%$). Consequently, a random-effects model was utilized to estimate the overall prevalence of depression in heart failure patients in Ethiopia. Additionally, heterogeneity was further evaluated through subgroup analysis. Despite this, the level of heterogeneity remained high, as illustrated in figures three to five. To address this, a meta-regression analysis was conducted, taking into account sample size and year of study. However, the variables analyzed in the meta-regression indicated that the heterogeneity in depression prevalence in Ethiopia was not linked to variations in sample size or study year (P = 0.871 and P = 0.752, respectively (Table 2). This considerable heterogeneity suggests that there are significant differences in the study outcomes that cannot be attributed solely to random chance.

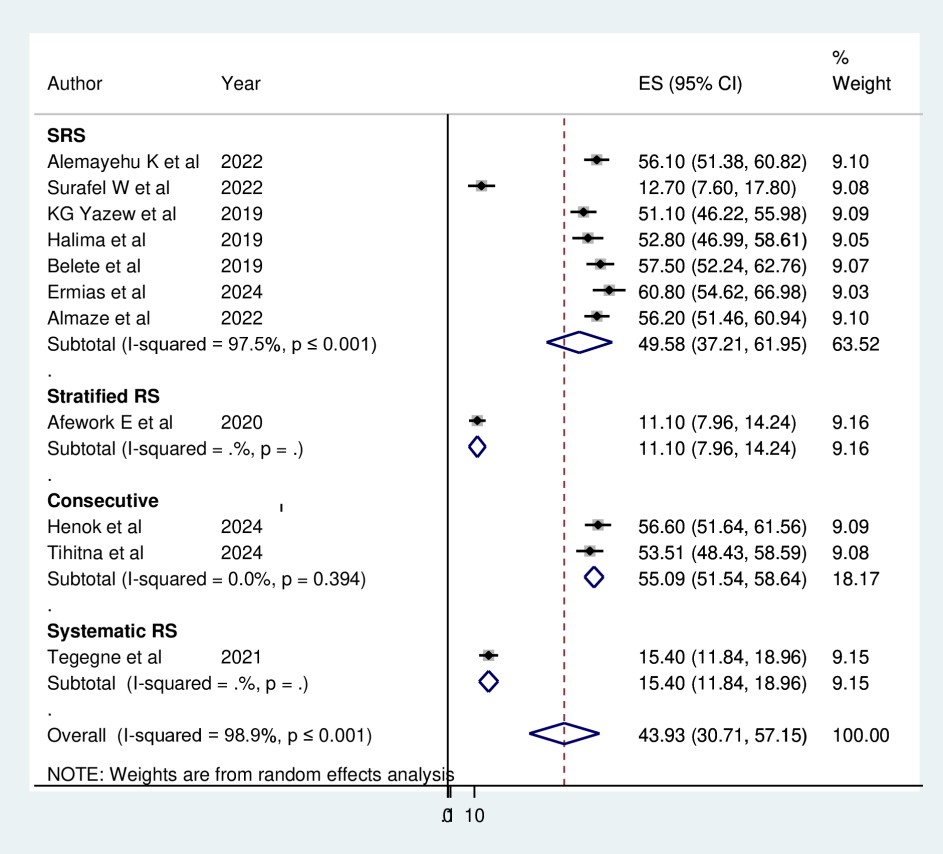

**Fig 5. Subgroup analysis by sampling technique on the prevalence of Depression among Heart failure patients in Ethiopia, 2024.**

**Table 2. Meta-regression analysis of factors affecting between study heterogeneity, 2024.**

| Logp | Coef. | Std. Err. | t | P>|t| | [95% Conf. Interval] |
|---|---|---|---|---|---|
| Year | .1338904 | .4088881 | 0.33 | 0.752 | −.8090072 1.076788 |
| Sample size | .0016748 | .0099774 | 0.17 | 0.871 | -.021333 .0246827 |
| _cons | −267.8303 | 826.8911 | −0.32 | 0.754 | −2174.645 1638.984 |

**Funnel plot of the risk of publication bias for Depression among HF patients.** Publication bias was assessed through the use of funnel plot and Egger's regression test statistics [36]. Qualitatively, the funnel plot was asymmetric which supporting the presence of publication bias by visual inspection as displayed in figure the three studies lie at the right side and eight studies appear at the left (Fig 6). Thus, it suggested that there was publication bias. Additionally, the Egger's weighted regression tests revealed indication of publication bias (p = 0.003) respectively (Table 3). To reduce this publication bias, trim and fill analysis was conducted (Fig 7).

**Sensitivity analysis.** A sensitivity analysis was conducted to assess the impact of each individual study on the estimated pooled prevalence of depression in this systematic review. The findings showed that no single study significantly affected the overall estimate of depression among individuals with heart failure (Fig 8).

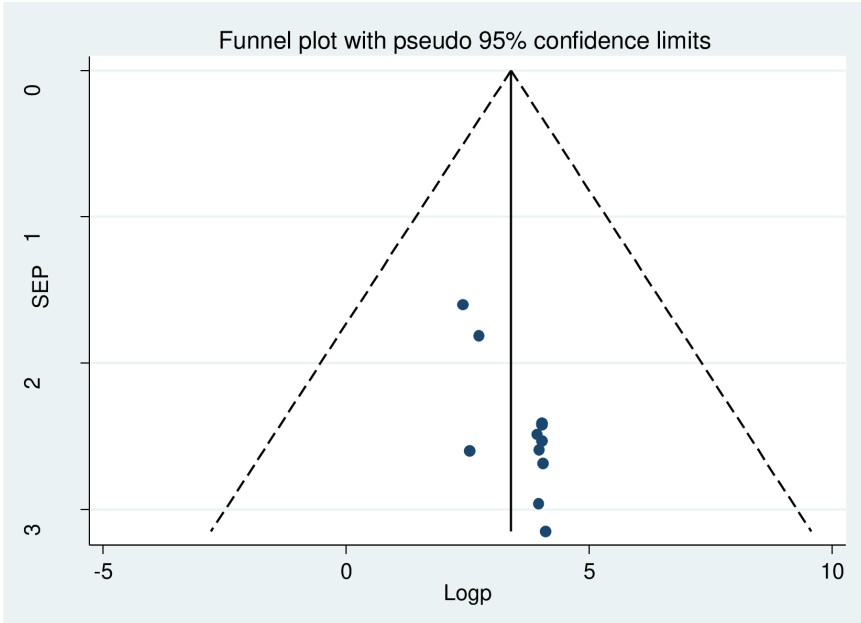

**Fig 6. Funnel plots to test the publication bias of the 11 studies, 2024.**

**Table 3. Publication bias of the Depression among Heart Failures in Ethiopia, 2024.**

| Std_Eff | Coef. | Std. Err | t | P>t | [95% Conf. Interval] |
|---|---|---|---|---|---|
| slope | .5514244 | .7147507 | 0.77 | 0.460 | −1.065454 2.168303 |
| bias | 1.240594 | .305007 | 4.07 | 0.003 | .5506207 1.930568 |

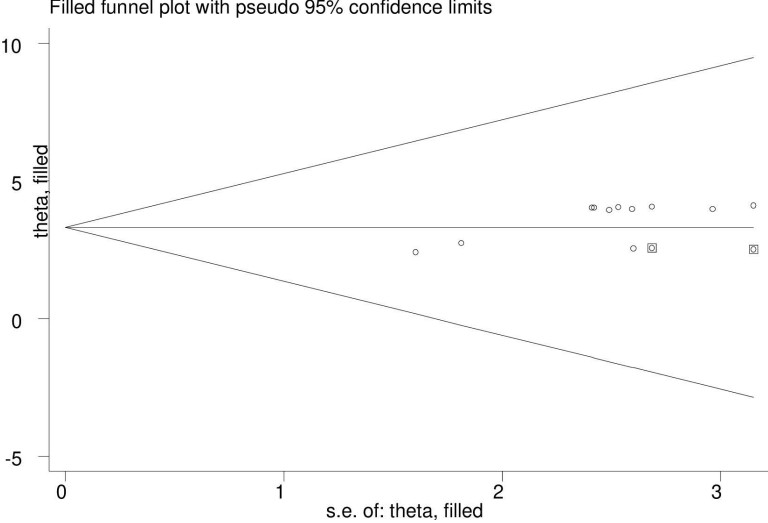

**Fig 7. Funnel plot shows trim and fill analysis depression among Heart failure patients in Ethiopia, 2024.**

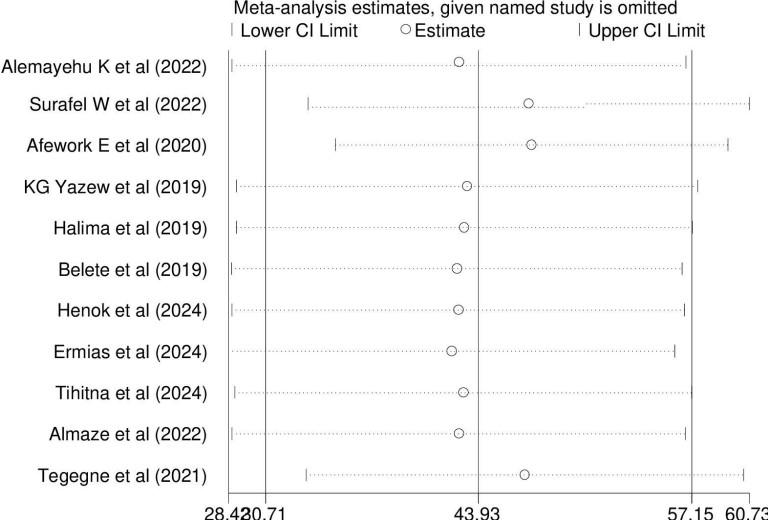

**Fig 8. Sensitivity analysis of depression among Heart failure in Ethiopia, 2024.**

## Discussion

Heart failure represents the final stage in various heart diseases [37] and is linked to significant morbidity and mortality rates. This has a notable impact on the healthcare system as patients often require repeated hospital admissions, and it also has a substantial effect on mental well-being [38,39]. This study synthesized the results of eleven primary investigations performed among Heart failure patients in Ethiopia, to determine the pooled prevalence depression involving 3807 study participants.

A systematic review showed that pooled depression approximately 43.93% with a 95% CI (30.24, 57.15) of heart failure patients experience symptoms of depression, a statistic consistent with studies in low-income countries (56.7%) [40], China (43%) [41], and Iran (47%) [42]. However, the combined estimate in this review shows a higher prevalence of depressive symptoms among Ethiopian heart failure patients compared to studies in Australia (15%) [43], Japan (5.8%) [44], and among black Americans (20.3%) [45]. The possible explanation for these differences might be attributed to factors such as poor adherence to self-care behaviors in Ethiopia [46], differences in study populations, the educational systems, and setting variations in resource allocation as well as continued professional development.

Despite this, some studies have reported even higher rates of depression among heart failure patients [47–49]. Possible reasons for these discrepancies include methodological issues, sample sizes, and socio-demographic characteristics. Although the ability to detect significant differences was limited due to analysis methods and a lack of studies, the risk of depression appears to be more common in heart failure patients compared to the general population [50].

The subgroup analysis revealed that depression rates among heart failure patients varied significantly by study area, with the highest prevalence of 60.80% recorded in SNNPR and the lowest at 35.18% in AA. This discrepancy may be attributed to the differences in participant numbers among the primary studies [51]. Despite their differing setups, the levels of depression among heart failure patients were found to be remarkably similar in both the Teaching Hospital and the Referral Hospital. This similarity may exist despite their differences, as both institutions suggested that the presence of medical residents and a higher patient volume for advanced procedures could affect the quality of care, potentially influencing depression rates.

Additionally, the analysis based on sampling techniques indicated that depression among heart failure patients was highest at 49.58% with simple random sampling and lowest at 11.1% with stratified random sampling. The differences in

these rates could be linked to variations in sample size [51] and the effects of different sampling methods on outcomes [52]. While many factors were explored in each study, conducting a factor analysis was not possible in this review due to significant heterogeneity and inconsistent factors across the studies.

## Conclusion and recommendations

The pooled prevalence of depression among patients with HF was high. The difference was observed in the prevalence of depression in a different region of the country. The most prevalence of depression was found in SNNPR compared to other regions. Health education given by health care providers at all levels might be treatment and self-care adherence issues. Therefore, this study will assist the Ethiopian policymakers and health personnel to focus on designing institutional and community based prevention and treatment of cardiac patients with depression. Plus, needs to work on enhancement of knowledge how to prevent, minimize, and treat the depression among HF patients as well as general population. Moreover, future research might recommended to focus on identification of the associated factors with depression among heart failure patients.

### Strength

This review has provided valuable information on the estimated pooled prevalence of depression among HF patients by using different databases to search the articles and by using six reviewers to minimize reviewers' bias and quality evaluation.

### Limitations

There are some limitations that could be addressed in future reviews. The first limitations of this study was the factor analysis was not conducted because of its high heterogeneity and lack of similar factors to analyze. Second, the included articles published only by the English language and all of them were cross-sectional, which might share the nature of cross-sectional study design limitations. Furthermore, publication bias and absence or scarce of published data from some regions of Ethiopia affects the subgroup analysis to interpret. Thus, may put to the difficulty to generalize the results.

## Supporting information

**S1 File. PRISMA 2020 Checklist.**
(DOCX)

**S2 File. AMSTAR checklist.**
(DOCX)

**S1 Table. Search strategy of databases up to 28th November 2024.**
(DOCX)

**S2 Table. JBI Critical appraisal checklist for eligible studies.**
(DOCX)

**S3 Table. List of excluded full texts with reasons for exclusion.**
(DOCX)

**S1 Data. Depression among HF.**
(XLSX)

## Acknowledgments

We would like to acknowledge all the authors of the original studies included in this systematic review.

## Author contributions

**Conceptualization:** Birhaneslasie Gebeyehu Yazew.

**Data curation:** Birhaneslasie Gebeyehu Yazew, Daniel Adane Endalew.

**Formal analysis:** Birhaneslasie Gebeyehu Yazew, Workineh Tamir Shitu, Zewdu Bishaw Aynalem.

**Investigation:** Birhaneslasie Gebeyehu Yazew.

**Methodology:** Birhaneslasie Gebeyehu Yazew, Workineh Tamir Shitu, Zewdu Bishaw Aynalem, Daniel Adane Endalew.

**Project administration:** Birhaneslasie Gebeyehu Yazew, Mohammed Hassen Salih, Haile Workye Agazhu.

**Resources:** Birhaneslasie Gebeyehu Yazew.

**Software:** Birhaneslasie Gebeyehu Yazew.

**Supervision:** Birhaneslasie Gebeyehu Yazew.

**Validation:** Birhaneslasie Gebeyehu Yazew.

**Visualization:** Birhaneslasie Gebeyehu Yazew.

**Writing – original draft:** Birhaneslasie Gebeyehu Yazew, Workineh Tamir Shitu, Mohammed Hassen Salih, Zewdu Bishaw Aynalem, Haile Workye Agazhu, Daniel Adane Endalew.

**Writing – review & editing:** Birhaneslasie Gebeyehu Yazew, Mohammed Hassen Salih, Zewdu Bishaw Aynalem, Haile Workye Agazhu, Daniel Adane Endalew.

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
