## [Decision Letter · Decision Letter 0]

Dear Dr. Yazew,

Thank you for submitting your manuscript to PLOS ONE. After careful consideration, we feel that it has merit but does not fully meet PLOS ONE’s publication criteria as it currently stands. Therefore, we invite you to submit a revised version of the manuscript that addresses the points raised during the review process.

**ACADEMIC EDITOR: **

The database search was conducted in March 2023, but the PROSPERO protocol has not been registered until now,  which is a significant limitation.Since that search,  relevant papers have been published on this topic and should be included in the analysis. You conducted  a meta-analysis of proportions using STATA version 11. However, this analysis type was introduced in late 2023, and while some researchers provided a (metaprop) command for this analysis in 2014, it only functioned on STATA version 14 or later. I am curious about how the authors managed to perform it with version 11.Check your PRISMA flow chart. It should be updated and recent one 

We look forward to receiving your revised manuscript.

Kind regards,

Adera Debella Kebede, MSC

Academic Editor

PLOS ONE

2. Please identify your study as "systematic review and meta-analysis"  in the title of your manuscript.

3. In the online submission form, you indicated that [Data will be available upon request of corresponding author].

6. Please include a separate caption for each figure in your manuscript.

7. As required by our policy on Data Availability, please ensure your manuscript or supplementary information includes the following:

Reviewers' comments:

Reviewer's Responses to Questions

**Comments to the Author**

1. Is the manuscript technically sound, and do the data support the conclusions?

Reviewer #1: Partly

Reviewer #2: No

Reviewer #3: Yes

2. Has the statistical analysis been performed appropriately and rigorously?

Reviewer #1: No

Reviewer #2: No

Reviewer #3: Yes

3. Have the authors made all data underlying the findings in their manuscript fully available?

Reviewer #1: Yes

Reviewer #2: No

Reviewer #3: Yes

4. Is the manuscript presented in an intelligible fashion and written in standard English?

Reviewer #1: Yes

Reviewer #2: No

Reviewer #3: Yes

Reviewer #1: Dear Editor and author,

I would like to extend my heartfelt thanks to Addisalem Workie, the academic editor of PLOS ONE, for inviting me to review the manuscript titled "A Systematic Review of Depression among Heart Failure Patients in Ethiopia, 2023" for PLOS ONE. It was an honor to be entrusted with this task.

Reviewing this manuscript was an enriching experience. The research fills a crucial gap in the literature and addresses a significant public health issue in Ethiopia. I commend the authors for their meticulous work, and I also commend the editorial team for recognizing its importance.

Thank you once again for this opportunity to contribute to the advancement of knowledge in this field. I appreciate the chance to engage with this important topic. However, I have identified several limitations within the study that need to be addressed before I can recommend acceptance for publication. Please carefully review the comments I have provided below:

1. Your topic is interesting, but there are already two papers published on it, both systematic reviews and meta-analyses. What makes your paper unique? If there are differences, please clearly state them to avoid redundancy. Please refer to the following publications and respond accordingly:

a. Mulugeta et al. Prevalence of depression and its association with health-related quality of life in people with heart failure in low- and middle-income countries: A systematic review and meta-analysis.

b. Firomsa et al. Adherence to self-care practices and associated factors among heart failure patients in Ethiopia: A systematic review and meta-analysis.

2. Regarding data availability, in your online submission, you stated/selected that "all data are fully available without restriction," but in your online submission, you wrote "Data will be available upon request of corresponding author." Please ensure consistency in your statements.

In the Methods section of your abstract, you mentioned, "Analysis was conducted using STATA version 11 software, with assessment of heterogeneity and publication bias. A random-effects meta-analysis model was used to estimate the pooled prevalence of depression." However, I could not find the assessment of heterogeneity and publication bias in the manuscript. Additionally, you uploaded the subgroup analysis of heterogeneity without discussing them in the Results section. Please revise your Results section to include all subgroup analysis results and their respective values. If there is heterogeneity, you have to conduct Univariate Meta-Regression and report your findings. Based on those results, you need to separately present your subgroup analysis findings.

3. Where is the statistical measure of your publication bias?

4. Under the Results section, there are no graphical representations of the included studies. Please update your manuscript to include appropriate tables, figures, or any other illustrative formats.

5. Your results section lacks chronological order, coherence, or may be missing information. Please revise it for clarity and organization.

6. There is evidence of copywriting or plagiarism

7. Your discussion section is lacking in depth and clarity. Please work on making it understandable, well-discussed, justified, and engaging.

8. Avoid using abbreviations in captions, titles, or abstracts.

9. Where are your recommendations??

10. Modify figure 1.

11. Your declaration section lacks coherence and completeness. For instance:

i. Your statement "Ethics approval and consent to participate, Acknowledgements: Not applicable" requires clarification. Why is it not applicable?

ii. Regarding the availability of data and materials: "All available data are within the paper" suggests full accessibility without restrictions, while "Data will be available upon request of the corresponding author" implies a different level of accessibility. There is inconsistency regarding data availability in both your manuscript and the online submission.

Reviewer #2: I would like to thank the authors for their efforts in analyzing the prevalence of depression among HF patients in Ethiopia. Unfortunately, given the numerous limitations in the manuscript, my response cannot be more positive.

Database search

1- The database search has been done in March 2023 and yet the PROSPERO protocol was not registered thus far. It means that you had it registered post-hoc, which is a major drawback.

2- Since March 2023, 2-3 relevant papers have been published on this topic that should be included in the analysis. I assume more would be found on other database (particularly Google Scholar). You need an updated search.

3- I am assuming there is a problem with the syntax you used for the database search because getting only 6 studies out of 150631 screened articles outlines a HUGE Problem, either with the syntax or with the screening. You would need to provide the detailed search query employed for each database. Also, please be advised that only the 1st 200 records from Google Scholar are to be included (as per guidelines).

Methodology:

1- Having 6 authors do a review of 6 papers is a bit confusing to me.

2- You have not outlined how depression and HF were diagnosed in each study. Maybe there could be a difference based on the diagnostic criteria.

3- Again, the limited sample size makes it very difficult to delve into the (effect modifying role) of these paramters (i.e., diagnositc criteria, sampling method, etc.) because you need to have at least 3 studies in each subgroup to provide interpretable evidence; otherwise, it would be too inaccurate.

4- You did not report the severity of depression +/- HF in the analysis.

5- The English structure of the paper is very poor. It needs professional editing.

Statistical analysis

1- The authors performed meta-analysis of proportion on STATA version 11. This type of analysis was introduced in late 2023 to STATA. However, some authors provided a (metaprop) command to carry out this analysis in 2014. At that time, this command would only run on STATA version 14 or beyond. I am wondering how the authors performed it using version 11.

2- Publication bias assessments needs at least 10 studies to make (powered analysis) through Funnel plot. Runnign a Funnel plot with 6 studies is not powered to detect any publication bias.

Added value:

1- There is an already published MA on this topic which provides not only region-based prevalence of Depression in HF, but also its imapct on quality of life. The paper is already published in PLOS ONE (https://pmc.ncbi.nlm.nih.gov/articles/PMC10035817/)

Reviewer #3: REVIWER COMMENT AND SUGGESTIONS

I would like to congratulate the Authors for raising this topic pertaining the Depression among Heart failure. When, I have been reading throughout the manuscript the idea was really interesting.

However, there are few comments and suggestions for authors for the aim of improving this paper.

THE TITLE

o The title is very narrow I suggest the authors to improve and make it to sound scientific manner.

ABSTRACT

METHODS

o I noted the authors didn’t address the study durations also inclusions criteria of the study.

RESULT

o On part of result I noted the authors didn’t explain some of information which is very important. I suggest the authors to revise and improve on it.

CONCLUSIONS

o The author should I make the conclusions according to the result.

INTRODUCTIONS

o Very clear and understanding but I noted the authors should correct at the last paragraph of introductions To the best of our knowledge, there is no existing systematic review on the prevalence of depressive symptoms among heart failure patients in Ethiopia. Understanding and estimating depression among heart failure patients can aid in enhancing the health status of individuals by addressing the negative impact of depression and other complications related to heart failure. Therefore, this systematic review aims to consolidate data on the prevalence of depressive symptoms among heart failure patients in Ethiopia. I suggest the authors should use the scientific language.

o I would ask the authors why the review is necessary gaps in existing of knowledge. I suggest make clear justifications of your review.

METHODS

o On part of study protocol and registrations I suggest this is systematic review manuscript so if it has no registrations number it won’t be publish its better to have them ready.

o I noted the authors written this A systematic literature search had made by two investigators (BG and YB) what does it meaning? I suggest to remove are not fit on this part already classify on another part.

o Also the authors should revise on this part there are some error appeared like (((((Depression [Mesh Terms]) OR (depress*[Mesh Terms])) OR (“Mental disorder” [Text Word])) OR (“Mood disorder” [Text Word])) AND ((Heart failure) OR (congestive heart failure [Mesh Terms])) I suggest improve them and make it clear.

Outcome measure

o The authors should explain each one and their procedure.

o Where is the diagram on part of study selection.it should be included on your document

o On this part the authors should be revise are not clear some of information is confusing when reading.

o TABLE 1 I noted the authors should revise because the way the authors present the heading above is not clear.

DISCUSSIONS

o In the part of discussion I noted is very narrow there is no comparison of the findings align or contract with prior research the authors should revise and improve them.

o What is your strength on your systematic review?

LIMITATIONS

o Also there are no limitations concerning to your study such as potential bias limitations in the available data and study limitations. I suggest the authors should revise and improve on it.

Declaration

o I adverse the authors to remove this part and follow PLOSE ONE guideline

NB:

o The authors should consider the spelling errors when writing the manuscript also the authors should revise all documents and use the high Standard English which is used in research. Because this paper are very interesting.

**Do you want your identity to be public for this peer review?** For information about this choice, including consent withdrawal, please see our Privacy Policy

Reviewer #1: No

Reviewer #2: No

Reviewer #3: No

---

## [Author Response · Author response to Decision Letter 1]

10 Jan 2025

Thank you for your valuable comment and questions!

All given comments are replied and attached by the table.

---

## [Decision Letter · Decision Letter 1]

Prevalence of the depression among Heart failure patients in Ethiopia, 2024: A Systematic Review and Meta- Analysis

PONE-D-24-09646R1

Dear Dr. Gebeyehu Yazew

We’re pleased to inform you that your manuscript has been judged scientifically suitable for publication and will be formally accepted for publication once it meets all outstanding technical requirements.

Kind regards,

Adera Debella Kebede, MSC

Academic Editor

PLOS ONE

Additional Editor Comments (optional):

Reviewers' comments:

Reviewer's Responses to Questions

**Comments to the Author**

Reviewer #1: All comments have been addressed

Reviewer #3: All comments have been addressed

2. Is the manuscript technically sound, and do the data support the conclusions?

Reviewer #1: Partly

Reviewer #3: Yes

3. Has the statistical analysis been performed appropriately and rigorously?

Reviewer #1: N/A

Reviewer #3: Yes

4. Have the authors made all data underlying the findings in their manuscript fully available?

Reviewer #1: Yes

Reviewer #3: (No Response)

5. Is the manuscript presented in an intelligible fashion and written in standard English?

Reviewer #1: Yes

Reviewer #3: Yes

Reviewer #1: (No Response)

Reviewer #3: REVIEWER COMMENT AND SUGGESTIONS

Overall, congratulations to the authors for resubmitting this manuscript. However, there are a few minor issues that need to be addressed.

Tittle: The title is quite explicit, but I observed that in line 4, there is no need to rewrite the title.

ABSTRACT

Background: it is unnecessary to include the design on a portion of the background; instead, you should condense the problem statement and the objective of your research..

Result: In the results section, the authors should begin with the findings instead of starting with demographic information.

Conclusion: there is no need to write the design of the study the authors should conclude according to the result.

Methods: The authors need to make revisions to this section, specifically line number 129.

Result: I observed that line number 198 is unnecessary, so please eliminate it.

**Do you want your identity to be public for this peer review?** For information about this choice, including consent withdrawal, please see our Privacy Policy

Reviewer #1: **Yes: ** Gemeda Wakgari Kitil

Reviewer #3: **Yes: ** rehema abdallah

---

## [Editor Report · Acceptance letter]

PONE-D-24-09646R1

PLOS ONE

Dear Dr. Yazew,

I'm pleased to inform you that your manuscript has been deemed suitable for publication in PLOS ONE. Congratulations! Your manuscript is now being handed over to our production team.

Kind regards,

on behalf of

Dr. Adera Debella Kebede

Academic Editor

PLOS ONE